# The Dutch Pregnancy Drug Register: Suitable to Study Paternal Drug Exposures?

**DOI:** 10.3390/ijerph20237107

**Published:** 2023-11-24

**Authors:** Annerose E. van der Mijle, Petra J. Woestenberg, Leanne J. Kosse, Eugène P. van Puijenbroek

**Affiliations:** 1Netherlands Pharmacovigilance Centre Lareb, 5237 MH Hertogenbosch, The Netherlands; p.woestenberg@lareb.nl (P.J.W.); l.kosse@lareb.nl (L.J.K.); e.vanpuijenbroek@lareb.nl (E.P.v.P.); 2Groningen Research Institute of Pharmacy, PharmacoTherapy—Epidemiology & Economics, University of Groningen, 9713 AV Groningen, The Netherlands

**Keywords:** paternal exposure, Dutch Pregnancy Register, pregnancy outcome, immunosuppressants, pregnancy, medication, infant health, birth defect, fertility, time to pregnancy

## Abstract

Paternal medication use around the time of conception is common, but information about its effects on pregnancy outcome and the health of the child is generally limited. The aim of this study is to examine the feasibility of studying paternal exposure in the Dutch Pregnancy Drug Register by using immunosuppressants as a proof of concept. In 113 of 15,959 pregnancies, long-term paternal immunosuppressant use was reported 3 months before conception. In total, 134 immunosuppressants were used. Pregnancy outcome was known for 54 cases and was in accordance with previous findings. Two spontaneous abortions, two premature births, six small for gestational age babies, and two major congenital malformations were reported. Time to pregnancy (TTP) was known for 9548 pregnancies, including 89 with paternal immunosuppressant use. TTP analysis did not show a difference in pregnancies with paternal immunosuppressant use compared to the control group. Moreover, the number of fertility treatments in the paternal immunosuppressant group was similar to the control group. In our opinion, it is feasible to use the Dutch Pregnancy Drug Register to study the effects of paternal exposure on pregnancy outcome. However, to study the potential effects on fertility, more information is needed, particularly since the beginning of pregnancy attempts.

## 1. Introduction

Most studies examining the safety of medication use before and during pregnancy focus on maternal medication use and little is known about the safety of paternal exposure to medicinal products. However, medication use by the father is also common. A study that analyzed prescription data showed that approximately one-third of fathers use medication in the six months before conception in Denmark and in the Netherlands [1]. In addition, a study in the Netherlands that analyzed pharmacy data showed that 73% of fathers used medication in the three months before conception and during pregnancy [2]. Currently, it is unclear if some of these paternal drug exposures may have adverse effects on pregnancy outcome and the offspring. For example, in 2016, there was some disagreement around the paternal use of mycophenolate mofetil [3]. The discussion was centered around a recommendation of the European Medicines Agency (EMA) to add a statement in the Summary of Product Characteristics (SmPC) that men should use contraception during treatment and for at least 90 days after cessation of treatment with this drug. This recommendation contradicted the relatively reassuring but limited human data available with paternal use of mycophenolate mofetil. It is known that certain immunosuppressive drugs can have teratogenic properties when used by women during pregnancy. In addition, in some animal studies, these drugs have been shown to have an effect on the reproductive outcome after paternal exposure [4,5]. This shows that there is a great need for more information regarding paternal medication exposure, especially to immunosuppressive drugs.

The total cycle of spermatogenesis is approximately 74 days [6]. Therefore, paternal medication use in the first three months prior to conception can potentially have adverse effects on pregnancy and the offspring. There are several proposed mechanisms. First, the drug can influence spermatogenesis, thereby having an effect on fertility, as can be reflected in the time to pregnancy (TTP). For most drugs, the effects on fertility are known to be reversible, but irreversible effects have been observed for some drugs, such as cyclophosphamide [7,8]. Second, a drug could also have a detrimental effect on the genetic material of the sperm cell, theoretically causing genetic abnormalities in the offspring [9]. Another mechanism is the direct transportation of a drug or its metabolites via seminal fluid to a pregnant woman or embryo [9]. However, so far, there is no evidence that drugs enter the semen in relevant amounts [10]. Although most human studies do not show an effect of paternal medication exposure, or specifically the immunosuppressants, on pregnancy outcome, information is generally limited [11,12].

To improve safety information regarding medication use during pregnancy, pregnancy registers can be used to collect information on drug exposure in the time period before and during pregnancy using questionnaires. The focus of these registers is mainly maternal exposure. Medication use by fathers-to-be is rarely monitored systematically. The aim of this study is to investigate the feasibility of using the Dutch Pregnancy Drug Register (a pregnancy registry focused on maternal exposure) to collect information on the possible effects of paternal exposure on pregnancy outcome (spontaneous abortion/fetal death, stillbirth, preterm birth, small for gestational age, and the presence of major congenital malformations) and TTP, using immunosuppressive drugs as a proof of concept.

## 2. Materials and Methods

### 2.1. Study Design

Data from the Dutch Pregnancy Drug Register were used. The design of the Dutch Pregnancy Drug Register and the validation studies performed have been described in detail by Vorstenbosch and colleagues [13]. Briefly, data collection through web-based questionnaires that were completed by pregnant women started on 1 April 2014 and is still ongoing. The register has a prospective non-interventional observational cohort design. All pregnant women in the Netherlands aged 18 or above are allowed to participate. Women can participate with multiple pregnancies. Participants can enroll during the whole pregnancy. They receive questionnaires at enrollment, in gestational week 17 (if applicable) and 34, and 2, 6, and 12 months after the expected date of delivery. The questionnaires received during pregnancy include questions regarding maternal general health, lifestyle, and medication use. The first questionnaire also includes questions about long-term medication use by the father in the three months before conception (i.e., medication used daily, weekly, or monthly for a longer period of time). In addition, a question is asked about the time it took to become pregnant (categorized as one month, within a year, or more than one year). The questionnaires after delivery contain questions on pregnancy outcome, childbirth, and child health. All participants provided informed consent before participation.

### 2.2. Study Population

We included data for pregnant women who enrolled in the Dutch Pregnancy Drug Register between 1 April 2014 and 31 August 2022. The TTP question was implemented in the questionnaires since 14 January 2021. For the TTP analysis, all pregnant women who enrolled after the implementation were included. For the outcome analysis, all women with an estimated date of delivery before 31 May 2022 were included, so all participants had the chance to complete the first questionnaire postpartum.

### 2.3. Exposure Definition

We defined exposure as paternal immunosuppressant use in the three months before conception, including alkylating agents, aminoquinolines, aminosalicylic acid and similar agents, calcineurin inhibitors, CD20 (Clusters of Differentiation 20) inhibitors, interleukin inhibitors, protein kinase inhibitors, purine analogues, retinoids for cancer treatment, selective immunosuppressants, and tumor necrosis factor alpha (TNF-α) inhibitors. Corresponding Anatomical Therapeutic Chemical (ATC) codes for these drug classes are shown in the Appendix A.

The non-exposed (comparator) group contains all women in the Dutch Pregnancy Drug Register without paternal exposure with one of the above ATC codes.

### 2.4. Outcome Definition

For TTP analysis, women were classified into two groups: pregnant within a year or more than one year since the first pregnancy attempt. We also determined if the pregnancy was spontaneous or if assisted reproductive technology/fertility treatment was needed and whether this was because of the impaired fertility of the father.

Pregnancy outcomes of interest were spontaneous abortion (pregnancy loss prior to 20 weeks’ gestation), fetal death (≥20 weeks’ gestation), induced abortion, stillbirth, preterm birth (infant born before 37 weeks’ gestation), small for gestational age, and the presence of major congenital malformations. Small for gestational age was defined as a birthweight below the 10th percentile compared to newborns of the same gestational age according to the Perinatal Registration of Dutch Newborns (Perined, Utrecht, The Netherlands) birthweight curves [14]. Congenital malformations were classified as major according to the European Surveillance of Congenital Anomalies (EUROCAT) classification system [15].

### 2.5. Statistical Analysis

For the TTP analysis, generalized estimating equation (GEE) logistic regression was performed to account for multiple pregnancies for the same women. Women with an unplanned pregnancy or with an unknown TTP were excluded. Analysis was corrected for maternal and paternal age, country of birth of both father- and mother-to-be, education level of both father- and mother-to-be, maternal BMI, and gravidity. Differences in fertility treatments performed in order to conceive and causes for the medical treatment were compared using a Chi-square test and Fisher’s exact test. In cases where it was not known whether the woman had undergone a fertility treatment, they were excluded.

To describe the rate of preterm birth and small for gestational age, and the number and characteristics of congenital malformations, a case series was performed for the cases with paternal immunosuppressant use and abnormal pregnancy outcome. The following information was collected: demographic factors, maternal medication use, maternal disorders, pregnancy complications, pregnancy outcome, and information on the health of the infant (i.e., birth weight, prematurity, congenital malformations).

All statistical analyses were performed using R studio version 4.1.3, with a statistical significance level of *p* < 0.05.

## 3. Results

In the pregnancy register, information about the father was collected in the first questionnaire. We collected information on whether the biological father was known, the birth year, birth country, and education level. Also, information was obtained on whether the father used long-term medication in the three months before conception, and if so, what medication and for what indication.

A total of 15,959 pregnancies were registered in the pregnancy register on 31 August 2022 with a completed first questionnaire. In 1793 (11.2%) pregnancies, long-term paternal medication use in the three months before conception was reported. In 113 of those pregnancies, a total of 134 immunosuppressive drugs were used by the father. The 10 most commonly used immunosuppressive drugs are shown in Table 1.

### 3.1. Time to Pregnancy

A question on the TTP was posed in 10,429 pregnancies. In 826 cases, the pregnancy was unplanned and for 55 pregnancies, the TTP was unknown, leaving 89 pregnancies with paternal immunosuppressive drug use and 9459 pregnancies with no paternal immunosuppressive drug use (Appendix A). TTP was longer than a year in 13.5% of the pregnancies with paternal immunosuppressant use, compared to 15.8% in the control group. This difference was not statistically significant (*p* = 0.55). Moreover, no discrepancy was found after correcting for age, birth country and education level of both father and mother, and BMI and gravidity (*p* = 0.66).

### 3.2. Fertility Treatment

In 9% of pregnancies with paternal immunosuppressant exposure, a fertility treatment had taken place, compared to 9.6% in the control group (*p* = 0.84) (Appendix A). Of these pregnancies, reduced fertility of the men was mentioned as the cause for the fertility treatment in 3 (37.5%) and 232 (25.5%) of the pregnancies with and without paternal immunosuppressant exposure, respectively (*p* = 0.43).

### 3.3. Pregnancy Outcome

Pregnancy outcome was known for 54 pregnancies with paternal immunosuppressant exposure. A total of 52 children were born. Two pregnancies ended in a spontaneous abortion. Two children were born prematurely, six children were small for gestational age, and two major congenital malformations were reported, with a congenital malformation rate of 3.3% (95% CI: 0.9–11.19). The congenital malformations concerned a ventricular septum defect (VSD) and hydronephrosis. More detailed information of the cases with abnormal outcomes is shown in Table 2.

## 4. Discussion

We examined the feasibility of studying paternal medication exposure in the Dutch Pregnancy Drug Register in relation to information reported by the mother on pregnancy outcome and TTP.

We have uncovered key reasons why, to study the possible effects of paternal medication use on fertility, the pregnancy register is not suitable. An important limitation of the use of the pregnancy register for data collection is that one of the criteria for participation is a confirmed pregnancy. Women who have not been able to conceive cannot participate. Paternal medication use that influences fertility and thus does not lead to a pregnancy will therefore be missed in the analysis. In addition, the pregnancy register only contains (retrospective) information on chronic medication use in the three months before conception, while the TTP could have been longer than a year. Medication use before the three-month period is unknown. It is preferred to obtain prospective information from the beginning of pregnancy attempts, but that is not feasible, given how our pregnancy registry is set up. Although TTP analysis in our study did not show a difference in pregnancies with paternal immunosuppressant use compared to the control group, other studies have shown a possible effect on sperm quality with some immunosuppressive drugs [11]. A possible effect on TTP cannot be ruled out.

However, despite, those negatives, because information on paternal exposure and pregnancy outcome can be collected in the register, we believe that it is feasible to study paternal exposure and the possible effects on pregnancy outcome using the pregnancy register. The numbers in our study were small, and the data did not show an increased risk of adverse pregnancy outcome after paternal use of immunosuppressant drugs in the three months before conception. We propose that larger numbers are needed to demonstrate a possible association. The congenital malformations rate of 3.3% (95% CI: 0.9–11.19) is comparable with the prevalence of around 2.5% according to EUROCAT, the European network of population-based registries for the epidemiological surveillance of congenital anomalies [16]. The observed frequencies of preterm birth (3.7%) and small for gestational age (11.1%) were comparable with the rates reported for the general population in the Netherlands (7% and 10%, respectively) in 2021 [17]. Our results on pregnancy outcome are in accordance with previous findings on paternal immunosuppressant use [11,12].

A major advantage of the use of a pregnancy register to study paternal exposure is that similar data sources exist in many countries. A lot of research has already been conducted regarding maternal medication use during pregnancy [18], which is the primary purpose of the pregnancy register. By adding a few questions about the father (i.e., demographic information, medication use in the three months prior to pregnancy), like in the Dutch Pregnancy Drug Register, paternal exposure can also be considered.

A limitation of the use of the Dutch Pregnancy Drug Register is that the available information about the father is currently minimal, because the pregnancy register is designed from the perspective of the mother. All information concerning the father is reported by the mother and is therefore prone to errors. Moreover, information on paternal alcohol consumption, smoking, and illicit drug use is missing, given the registry’s primary focus on women. Also, occasional medication use by the father and comorbidities are unknown in the three months before conception. Information on the severity of the father’s condition that is being treated through the use of the drug is also missing; therefore, confounding by indication cannot be ruled out.

To make the pregnancy registry even more suitable for the examination of the effects of paternal exposure on pregnancy outcome, additional questions can be added to the questionnaires. For example, questions regarding the severity of the disease, comorbidities, alcohol, smoking, and illicit drug use can be included. However, it is not desirable to make the questionnaires too extensive for the participants because then it becomes too much of a burden to participate. This should therefore always be an important consideration. In addition, it is important to take country-specific privacy legislation into account when information about the father is reported by the mother.

To examine the possible effects of paternal drug use on fertility, it would be valuable to have information about semen parameters. Semen analysis, i.e., semen volume, sperm concentration, motility, and morphology, can be useful in the evaluation of fertility [19]. However, this information will often be unknown by the participants, and it is not possible to collect this information for this type of research.

Another possibility to study paternal exposure is to conduct research on the fathers themselves. However, the current privacy regulations in the Netherlands require the mother to provide consent for the father to report information on the health of his child. This complicates the study design and inclusion of patients. Moreover, if the study solely focuses on paternal exposure, valuable information on maternal medication use and lifestyle will be missing. Therefore, it is preferable to obtain sufficient information on both the father and mother. To study the potential effects of paternal exposure on fertility, it is preferable to follow couples from the start of pregnancy attempts, although this may be challenging to operationalize. Other options for data collection are the use and linkage of existing databases with information on prescription data, pregnancy outcome, and mother-father linkage. However, there are several other limitations on the usage of this type of information [20].

## 5. Conclusions

In conclusion, if detailed information on paternal exposure and possible confounders can be collected, it is feasible to study the effects of paternal medication exposure on pregnancy outcome with the use of a pregnancy register. However, to study the potential effects on fertility, more detailed information is needed from the father starting from the onset of pregnancy attempts, limiting the feasibility of using a pregnancy register.

## Figures and Tables

**Table 1 ijerph-20-07107-t001:** Most commonly reported paternal immunosuppressive drugs used in the three months before conception.

Drug	N	%
Mesalazine	36	26.9%
Azathioprine	15	11.2%
Adalimumab	15	11.2%
Infliximab	10	7.5%
Ustekinumab	7	5.2%
Mycophenolic acid/ Mycophenolate mofetil	7	5.2%
Tioguanine	6	4.5%
Tacrolimus (systemic)	6	4.5%
Tacrolimus (topical)	5	3.7%
Hydroxychloroquine	5	3.7%

**Table 2 ijerph-20-07107-t002:** Detailed description of the cases with abnormal outcomes. GA: gestational age; OHSS: ovarian hyperstimulation syndrome; PCOS: polycystic ovarian syndrome, SGA: small for gestational age; VSD: ventricular septum defect; SGA: small for gestational age; na: not applicable.

Case	Paternal Medication Use	Paternal Indication	Paternal Co-Medication	Maternal Age	Paternal Age	Maternal Comorbidity	Maternal Medication Use	Pregnancy Complications	GA at Time of Delivery	Birth Weight (Gram)	Pregnancy Outcome
1	Tacrolimus, mycophenolate mofetil	Kidney transplant	Metoprolol, cetirizine, candesartan, nifedipine, prednisolone, omeprazole	30–35	30–35	-	COVID-19 vaccine	-	na	na	Miscarriage at 7 weeks
2	Mesalazine	Crohn’s disease	-	30–35	30–35	PCOS	COVID-19 vaccine	-	na	na	Miscarriage at 12 weeks
3	Adalimumab	Rheumatoid arthritis	-	35–40	35–40	Hay fever, allergy	Miconazole, zinc sulfate, cefaclor, pertussis vaccine	-	40	3570	Congenital malformation: Hydronephrosis
4	Dupilumab	Eczema	Levocetirizine, budesonide/formoterol, mometasone	25–30	30–35	Irritable bowel syndrome	Acetaminophen, salicylic acid, pertussis vaccine, COVID-19 vaccine	Gestational hypertension	39	3770	Congenital malformation: VSD
5	Infliximab	Crohn’s disease	-	30–35	30–35	Anxiety	Meclozine/pyridoxine, acetaminophen, pertussis vaccine	-	35	2545	Premature
6	Mycophenolate mofetil, tacrolimus	Kidney transplant	Esomeprazole	30–35	30–35	Hay fever, fibromyalgia	Loratadine, fluticasone, COVID-19 vaccine, tramadol, pertussis vaccine	OHSS	34	2760	Premature
7	Adalimumab, hydroxychloroquine	Rheumatic disease	-	35–40	40–45	Ulcerative colitis	Meclozine/pyridoxin, metoclopramide, acetaminophen, macrogol, COVID-19 vaccine, calcium carbonate/magnesium, pertussis vaccine	-	40	3000	SGA
8	Infliximab	Ulcerative colitis	-	25–30	35–40	-	Nitrofurantoin, pertussis vaccine, COVID-19 vaccine	Gestational hypertension, pre-eclampsia	40	2889	SGA
9	Azathioprine	Cardiac sarcoidosis	Prednisone	40–45	40–45	Ankylosing spondylitis	Pertussis vaccine, acetylsalicylic acid/acetaminophen, COVID-19 vaccine	-	40	3061	SGA
10	Azathioprine	Crohn’s disease	-	30–35	30–35	Hashimoto′s disease, migraine	Levothyroxine, sumatriptan, clotrimazole, COVID-19 vaccine, pertussis vaccine, acetaminophen	-	40	3200	SGA
11	Adalimumab	Rheumatic disease	-	30–35	45–50	Epilepsy	Lamotrigine	-	40	2960	SGA
12	Mesalazine	Unknown	-	35–40	40–45	Epilepsy	Trachitol, acetaminophen, dabigatran, lamotrigine, fraxiparine, influenza vaccine, antagel, macrogol, pertussis vaccine	-	39	2700	SGA

## Data Availability

The data are not publicly available due to privacy and ethical restrictions.

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
