# Peer review of "The Dutch Pregnancy Drug Register: Suitable to Study Paternal Drug Exposures?"

_ijerph, 2023, doi:10.3390/ijerph20237107_

Round 1

Reviewer 1 Report

Comments and Suggestions for Authors

Review report for authors:

A brief summary:

The aim of this study was to examine the feasibility to study paternal exposures in the Dutch Pregnancy Drug Register, using immunosuppressants as a proof of concept.  The authors presented in this brief report  that TTP ( time-to-pregnancy) analysis did not show  a difference in pregnancies with paternal immunosuppressant use compared to the control group. The amount of fertility treatments was similar in the paternal immunosuppressant and  control group.

Like the authors said the aim of this study is to investigate the possibilities to use the Dutch Pregnancy Drug Register (a pregnancy registry focused on maternal exposure) to collect information on the possible effect of paternal exposures on pregnancy outcomes (spontaneous abortion/fetal death, stillbirth, preterm birth, small for gestational age (SGA) and the presence  of major congenital malformations) and TTP, using immunosuppressive drugs as a proof  of concept.

As the authors of this short review article has already been said a loto f research is already been done to maternal medications use during pregnancy, what the pregnancy register is primarly intended for. They said that by adding a relatively small amount of questions, paternal exposures can also be considered.

The paper is written in the form of a short review article extremely interesting to read, all the flaws of this research are clearly stated as well as the possibilities of refining it in future research.

The brief report is interesting both to perinatologists, as well as epidemiologists, human reproduction specialists, pediatricians, therefore a relatively wide population of specialists who deal with these medical problems.

General concept comments:

As the authors of this study already said, study has some limitations: the numbers in their study were small.  Another limitation that the authors pointed out is that the available information about the father is minimal, because the pregnancy register is designed for the perspective of the mother. All information concerning the father is reported by the mother, and thereby prone to errors. Moreover,  information on paternal alcohol consumption, smoking and drug use is missing, given the registry’s primary focus on women. Also occasional medication use by the father  and comorbidities are unknown in the three months before conception. Information on the severity of the condition the father uses the drug for is also missing.

These are limitations that you as the authors mentioned, but at the same time that are also clear limitations of the research.

The title is clear, well written and concise. It clearly points to the issue that will be discussed in the following text.

The summary is well written and contains all the important results and conculusions of the study.

I would like to add a few more keywords, there are only four.

The introduction  is clearly and interestingly written and provides an appropriate overview and introduction to the main topic of the paper.

The methods are examplary written and reproducible.

This paper is written in the form of a brief report so that the presentation of materials and methods is in full compliance with the general form of the article.

The results are adequately presented. The results include only tables, but the above tables meet this article form. The visual disadventage is the absence of graphs. Nevertheless, the results are easy to interpret and understand.

The discussion is clear and concise, the stages of the research are clearly stated, and this results are compared with the similar results in previosu similar findings.

 The statements in the discussion are drawn coherently and are supported by appropriate citations.

The literature is correctly cited.

 Slightly more than 35% of references  (35,29%, 6/17) are references of recent publications, which can perheps be explained by the relatively new and more comprehensive, so far not often applied concept of research.

I think that the work should be supported with more recent quotes.

The conclusion is consistent, concise with clearly presented arguments.

The hypothesis of the article is clearly written, explained and later developed methodologically. Experimental design of this study is appropriate for the testing  inicial hypothesis of the study.

In conclusion, I belive that this is an interesting work with a scientific contribution and it is applicable in a narrower, strictly defined medical population, and I recommend its acceptance for publication after minor corrections.

Author Response

Thank you very much for your feedback on our manuscript. We have studied the comments and revised the manuscript accordingly. Our responses to the comments are presented in a point-by-point manner. Textual changes to the manuscript are presented using track changes.

  1. I would like to add a few more keywords, there are only four.

Done as suggested.

  1. Slightly more than 35% of references (35,29%, 6/17) are references of recent publications, which can perheps be explained by the relatively new and more comprehensive, so far not often applied concept of research.
    I think that the work should be supported with more recent quotes.

We added more information on a quote (reference 3) in the introduction and added 4 more references in the discussion.

Reviewer 2 Report

Comments and Suggestions for Authors

Thank you for submitting this very interesting study about the use of the Dutch pregnancy registry and paternal drug exposure.

1. Line 31: Please add the word  “that” before “certain immunosuppressants”.

2. Line: 182: please explain to the reader what the EUROCAT is.

3. Lines 223-224: the statement about the information on maternal medication use is not clear and requires rephrasing to make it easily understandable to the reader. 

Author Response

Thank you very much for your feedback on our manuscript. We have studied the comments and revised the manuscript accordingly. Our responses to the comments are presented in a point-by-point manner. Textual changes to the manuscript are presented using track changes.

  1. Line 31: Please add the word “that” before “certain immunosuppressants”.

Done as suggested.

  1. Line 182: please explain to the reader what the EUROCAT is.

Done as suggested.

  1. Lines 223-224: the statement about the information on maternal medication use is not clear and requires rephrasing to make it easily understandable to the reader.

We have changed the sentence into “Moreover, if the study solely focuses on paternal exposures, valuable information on maternal medication use and lifestyle will be missing”.

Reviewer 3 Report

Comments and Suggestions for Authors

The present study focus on a subset of pregnant population, but both in maternal and paternal aspect. The study questions the feasibility of accessing Dutch Pregnancy Register to study the effects  of paternal exposures on pregnancy outcomes. This study seems to ongoing and updating the repository of information collecting from pregnant women and their partners.

1. Study seems to be interesting, however data collection can be modified by including the paternal age, history of paternal family for abortions, infertility etc. Authors discussed the importance of including the paternal smoking and drinking habits, which will give more information to correlate with the pregnancy outcomes. However, including the clinical parameters like testosterone levels, sperm count, other co-morbidities in the subjects involved in the study would be beneficial.

2. Study can be classified as a preliminary observation/approach for the possibility of accessing the registry for more conclusive results.

Author Response

Thank you very much for your feedback on our manuscript. We have studied the comments and revised the manuscript accordingly. Our responses to the comments are presented in a point-by-point manner. Textual changes to the manuscript are presented using track changes.

  1. Study seems to be interesting, however data collection can be modified by including the paternal age, history of paternal family for abortions, infertility etc. Authors discussed the importance of including the paternal smoking and drinking habits, which will give more information to correlate with the pregnancy outcomes. However, including the clinical parameters like testosterone levels, sperm count, other co-morbidities in the subjects involved in the study would be beneficial.

We added a column with paternal age in table 2. We already corrected for paternal age in the TTP analysis but this was not mentioned in the method section, so we added this as well in section 2.5. Unfortunately, we do not have information on history of paternal family for abortions or infertility. Additional questions to the questionnaires could be added, which we wrote a paragraph about in the discussion. It would indeed be useful to have information on semen parameters. We added a paragraph about this in the discussion.

  1. Study can be classified as a preliminary observation/approach for the possibility of accessing the registry for more conclusive results.

This is correct. The study focusses on “We examined the feasibility to study paternal medication exposures in the Dutch Pregnancy Drug Register” as mentioned now in the discussion section.

Reviewer 4 Report

Comments and Suggestions for Authors

1-     It is unclear whether the researchers in this study used the same questionnaires that were used in the aforementioned study No. 13, or if it is the same study and the data was gathered simultaneously, given that the two studies (the present and the referenced study No. 13) started at the same time on April 1, 2014.

2-     Table 1 need to write N

3-     It is not beneficial to make the questionnaires too detailed for the participants because then it becomes excessively difficult to participate, the authors stated in line 216 of their discussion after mentioning several limitations (lines 190, 200) and suggesting the addition of additional questions to collect more data.

I believe this is confusion, therefore what recommendations do the authors have to improve the tools both more informative and suitable for participants?

Author Response

Thank you very much for your feedback on our manuscript. We have studied the comments and revised the manuscript accordingly. Our responses to the comments are presented in a point-by-point manner. Textual changes to the manuscript are presented using track changes.

  1. It is unclear whether the researchers in this study used the same questionnaires that were used in the aforementioned study No. 13, or if it is the same study and the data was gathered simultaneously, given that the two studies (the present and the referenced study No. 13) started at the same time on April 1, 2014.

The study mentioned (reference 13) is the same study. We have moved the reference up so that it is clearer that it concerns the same study.

  1. Table 1 need to write N

Done as suggested.

  1. It is not beneficial to make the questionnaires too detailed for the participants because then it becomes excessively difficult to participate, the authors stated in line 216 of their discussion after mentioning several limitations (lines 190, 200) and suggesting the addition of additional questions to collect more data.
    I believe this is confusion, therefore what recommendations do the authors have to improve the tools both more informative and suitable for participants?

It is indeed contradictory that we say the registry is not suitable to study fertility and then we recommend adding additional questions to make it more suitable. These additional questions will not solve the most important limitation of the usage of the pregnancy register to study fertility, the lack of participants who did not become pregnant. We have corrected this in the discussion by focusing on pregnancy outcomes in this paragraph.

Reviewer 5 Report

Comments and Suggestions for Authors

The Title of the Study: The Dutch Pregnancy Drug Register: Suitable to Study Paternal Drug Exposures? is interesting and gives good insight about th content of the article.

The highlights the need for information on paternal medication exposures during pregnancy and explores the feasibility of using the Dutch Pregnancy Drug Register to study the effects of paternal exposures, focusing on immunosuppressants. The findings indicate that long-term paternal immunosuppressant use did not significantly impact pregnancy outcomes or fertility. However, more information is required to study the potential effects on fertility from the start of pregnancy planning.

Abstract: The abstract effectively summarizes the study's purpose and key findings.

Introduction: The introduction provides a comprehensive background of the research topic. It emphasizes the limited understanding of the safety implications of paternal medication use on pregnancy outcomes and offspring. The objectives are clearly stated. 

Methods: The authors used the Dutch Pregnancy Drug Register, which is a prospective observational cohort focused on maternal exposure. Questionnaires were distributed to pregnant women, collecting information on medication use by both mothers and fathers. This is a limitation of the study. Data collection could be biased. More information about the questionnaire and its validation is needed. 

The study population included pregnant women enrolled in the register between April 2014 and August 2022. Who answered the questionnaire?

Why the study focused on specific medication? 

Results: The results section presents the findings obtained from the analysis of the Dutch Pregnancy Drug Register data. The authors identified 113 pregnancies involving long-term paternal immunosuppressant use before conception. The results did not show a significant difference in pregnancy outcomes or TTP between pregnancies with paternal immunosuppressant use and the control group. 

The study showed that evaluating parental exposure is feasible; however, the effect of paternal exposure can't be evaluated with this number and this study design.

Discussion: The discussion section interprets the study's findings and relates them to the existing literature. The authors acknowledge the limitations of the study, such as the small sample size and potential underreporting of medication use. They highlight the need for further research on the effects of paternal medication exposures on fertility. The discussion should focus on the feasibility. This is the study objective. 

Conclusion: The conclusion effectively summarizes the main findings and emphasizes the potential of the Dutch Pregnancy Drug Register to study the effects of paternal exposures on pregnancy outcomes. It reiterates the need for more information on the effects of paternal medication use on fertility. The conclusion aligns with the study objectives and findings.

Reference list is accepted. 

Comments on the Quality of English Language

Minor revision

Author Response

Thank you very much for your feedback on our manuscript. We have studied the comments and revised the manuscript accordingly. Our responses to the comments are presented in a point-by-point manner. Textual changes to the manuscript are presented using track changes.

  1. Methods: The authors used the Dutch Pregnancy Drug Register, which is a prospective observational cohort focused on maternal exposure. Questionnaires were distributed to pregnant women, collecting information on medication use by both mothers and fathers. This is a limitation of the study. Data collection could be biased.

You correctly noted that this is an important limitation of our study. We describe this limitation in the discussion.

  1. More information about the questionnaire and its validation is needed.
    We added a sentence where we refer to reference 13 for more information on the study and performed validation studies.

  1. The study population included pregnant women enrolled in the register between April 2014 and August 2022. Who answered the questionnaire?

We added a sentence in 2.1 of the method section to clarify that pregnant women complete the questionnaires.

  1. Why the study focused on specific medication?

We describe this in the introduction. We have adjusted the order of the introduction to clarify why we focus on immunosuppressants.

  1. The study showed that evaluating parental exposure is feasible; however, the effect of paternal exposure can't be evaluated with this number and this study design.

This is correct. Larger numbers are needed, which we wrote about in the discussion.

  1. The discussion should focus on the feasibility. This is the study objective.

We have adjusted the discussion to focus more on feasibility. We also adjusted the order of the discussion to make the focus on feasibility clearer.

Reviewer 6 Report

Comments and Suggestions for Authors

Dear Authors,

Your "Brief Report" entitled "The Dutch Pregnancy Drug Register: suitable to study paternal drug exposures?" has been reviewed,

This paper deserves attention since it highlights an important topic related to the study of paternal drug exposures on pregnancy and delivery based on data collected from the Dutch pregnancy drug register.

Kindly find below my remarks and comments regarding your paper:

01- Regarding the Abstract section, This section is not clear at all for readers, it does not include sections (Background, Methods, Results, Conclusion), Authors are invited to paraphrase these sections to be clearer for readers.

02- In the Abstract section, Line 17, Authors are invited to put the full words related to TTP followed by (TTP).

03- In the Keywords section, Authors are invited to add the word "Dutch" to this list.

04- In the Introduction section, Line 26, Authors are invited to start the sentence with another word than "Although".

05- In the whole Manuscript, Authors are invited to use value (letter p should be in capital and in italic).

06- In the Results section, sub-section 3.1 "Time-to-pregnancy" Results in this paragraph are not shown in any table or figure.

07- In the Results section, sub-section 3.2 "Fertility treatment" Results in this paragraph are not shown in any table or figure.

08- In the Discussion section, just in the first paragraph authors used references, all other paragraphs do not contain references, noting that authors must discuss the results of this work with results of other studies.

09- The conclusion section is short and does not contain a lot of information.

Best Regards,

Author Response

Thank you very much for your feedback on our manuscript. We have studied the comments and revised the manuscript accordingly. Our responses to the comments are presented in a point-by-point manner. Textual changes to the manuscript are presented using track changes.

  1. Regarding the Abstract section, This section is not clear at all for readers, it does not include sections (Background, Methods, Results, Conclusion), Authors are invited to paraphrase these sections to be clearer for readers.

Unfortunately it is not possible to add headings to the abstract, according to the guidelines for authors of his journal.

  1. In the Abstract section, Line 17, Authors are invited to put the full words related to TTP followed by (TTP).

Done as suggested.

  1. In the Keywords section, Authors are invited to add the word "Dutch" to this list.

Done as suggested.

  1. In the Introduction section, Line 26, Authors are invited to start the sentence with another word than "Although".

Done as suggested.

  1. In the whole Manuscript, Authors are invited to use P value (letter p should be in capital and in italic).

Done as suggested.

  1. In the Results section, sub-section 3.1 "Time-to-pregnancy" Results in this paragraph are not shown in any table or figure.

We decided not to show these results in a figure or table. We prefer to describe these results in the text since the information is limited.

  1. In the Results section, sub-section 3.2 "Fertility treatment" Results in this paragraph are not shown in any table or figure.

Like the previous point, we decided not to show these results in a figure or table. We prefer to describe these results in the text since the information is limited.

  1. In the Discussion section, just in the first paragraph authors used references, all other paragraphs do not contain references, noting that authors must discuss the results of this work with results of other studies.

Thank you for this valuable comment, we added 4 more references to the discussion, allowing to place our findings in a broader context.

  1. The conclusion section is short and does not contain a lot of information.

We have expanded the conclusion a little. In our opinion a brief conclusion is more clear for our brief report.

Round 2

Reviewer 3 Report

Comments and Suggestions for Authors

Authors have addressed all the review questions satisfactorily. 

Author Response

Thank you for your response.

Reviewer 6 Report

Comments and Suggestions for Authors

Dear Authors,

Thank you for the modifications you made,

I still have the same remarks concerning my points number 6 and 7.

You are invited to put these results in additional or supplementary materials.

It is very important to show these results.

Best Regards,

Author Response

Thank you for your response. We added the tables with the results of the TTP analysis and the fertility treatment analysis in the supplementary materials.

Round 3

Reviewer 6 Report

Comments and Suggestions for Authors

Dear Authors,

Thank you for all the modifications you made,

BR,